# Rayleigh–Ritz Approximation of the Acoustic Vibrations of Clamped Superquadrics—Application to Free Core–Shell Objects

**DOI:** 10.3390/nano15241865

**Published:** 2025-12-11

**Authors:** Sajana S, María del Carmen Marco de Lucas, Lucien Saviot

**Affiliations:** Université Bourgogne Europe, CNRS, Laboratoire Interdisciplinaire Carnot de Bourgogne ICB UMR 6303, 21000 Dijon, France

**Keywords:** Rayleigh–Ritz method, acoustic vibration, core–shell, superquadrics, rigid boundary condition

## Abstract

A numerical approach based on the Rayleigh-Ritz method and using a modification of the so-called xyz algorithm is introduced to calculate the acoustic vibrations of clamped objects whose shape is delimited by superquadrics. It is then used to improve the convergence for the free vibrations of core–shell objects. The issue in this case is first illustrated in the simpler one-dimensional case of the thickness breathing vibration of an infinite “core-shell” plate. Functions suitable for solving the clamped vibrations of the core are added to the original xyz basis of functions to improve the convergence for core–shell superquadrics. The new basis obeys the same symmetry rules as the original one, which allows calculating vibrations for individual irreducible representations when the objects are made of cubic, tetragonal, or orthorhombic materials whose principal axes are aligned with those of the superquadrics. This method is validated for an isotropic spherical core–shell system for which analytic solutions exist.

## 1. Introduction

The acoustic eigenfrequencies of an object depend on its size, shape, the material it is made of, and its environment. Measuring and modeling these frequencies provides valuable information about all these parameters. Resonant ultrasound spectroscopy (RUS) implements this approach and is used mainly in geology and material science to determine the elastic constants of a material [1]. Fast numerical methods are required in this context to calculate the eigenfrequencies quickly enough to allow fitting the measured frequencies. The Rayleigh–Ritz (RR) method fulfills this requirement [2]. This method can be applied to objects with very different dimensions, down to nanomaterials. Different experimental methods allow measuring the eigenfrequencies of nano-objects in the time or frequency domains (for example, transient absorption and low-frequency Raman scattering) [3,4]. Recent experimental results in this domain have concerned more complex nano-objects in terms of shape and composition. In particular, gold–silver core–shell nanocrystals with spherical shapes have been synthesized and investigated with inelastic light scattering in order to assert the epitaxy (or lack thereof) between the gold core and the silver shell [5]. Gold–silver core–shell nanorods have also been synthesized [6] and their vibrations have been investigated with inelastic light scattering [7]. Their geometry is more complex because the silver coating is not homogeneous. The gold core has a circular cross-section, and the silver shell cross-section tends to a square with increasing thicknesses, and all these parameters must be taken into account to model their optical properties [6]. Superquadrics have been successfully used in this context to model the shapes of the core and the shell [8]. For these two core–shell systems, the gold core and the silver shell have a cubic lattice structure; therefore, anisotropic elasticity must also be taken into account.

Accurate modeling of such nano-objects is required to assign the different eigenfrequencies observed experimentally. The continuum elasticity approximation is valid in this context as long as the objects’ sizes are large enough, i.e., the wavelength of the elastic waves is large compared to the interatomic distance. The finite element method (FEM) is a very versatile method to compute eigenfrequencies for arbitrary shapes and elastic properties. However, in the previous examples, the 3D objects are invariant under many symmetry operations, which can be leveraged to help assign the experimental features as usually performed in spectroscopy. For example, transient absorption is essentially sensitive to eigenmodes for which a volume variation occurs during oscillation, i.e., the totally symmetric ones. Raman scattering also obeys selection rules preventing many eigenmodes from contributing to the spectra. Further processing of FEM calculations is generally required to recover the symmetry of the eigenmodes. RR can take advantage of symmetries to calculate only the eigenmodes of interest, i.e., those corresponding to specific irreducible representations. This also speeds up the calculations significantly and lowers the required computing resources.

The present work first extends to superquadrics a recently introduced method to calculate the vibrations of clamped spheres (rigid boundary condition, i.e., vanishing displacement at the surface) [9] by providing suitable basis functions meeting these requirements. This boundary condition is seldom encountered in practice, but it is a very useful limiting case, which helps figure out how the vibrations of objects embedded in a harder and/or denser matrix may differ compared to the free vibrations of the same object [10]. The vibrations of core–shell objects are then considered. They can be calculated using the usual RR method without modifications [5]. However, the convergence of the calculations is not always as good as for homogeneous objects, questioning the validity of this simple approach. This comes from the fact that the RR method is implemented with functions of class $C^{\infty }$, while the derivative of the displacement is generally different on both sides of an interface in core–shell systems. The present work proposes a method to improve the convergence using additional basis functions similar to those introduced before. This method is first demonstrated in the simpler 1D case of the thickness breathing vibrations of infinite plates. A calculation for a core–shell sphere is then presented and compared to the analytical solution to show the improved convergence.

## 2. Methods

RR calculations are based on the xyz algorithm introduced by Visscher et al. [2]. The main points are summarized below, and readers are referred to this paper for a comprehensive presentation. The three Cartesian components of the displacements are decomposed on the $\phi_{\lambda }=x^{p}y^{q}z^{r}$ basis of functions with *p*, *q*, and *r* integers, λ=(p,q,r), and p+q+r≤N. The RR approach turns the dynamic problem into the generalized eigenvalue problem $\omega^{2}Ea=\Gamma a$ with ω the pulsation, *a* the amplitudes of the $\phi_{\lambda }$ components, and *E* and Γ two square matrices defined as follows: (1)$$\begin{array}{ccc} E_{\lambda i\lambda^{\prime }i^{\prime }} & =\delta_{ii^{\prime }} & \int_{V}\phi_{\lambda }\rho \phi_{\lambda^{\prime }}dV \end{array}$$(2)$$\begin{array}{ccc} \Gamma_{\lambda i\lambda^{\prime }i^{\prime }} & =C_{iji^{\prime }j^{\prime }} & \int_{V}\phi_{\lambda ,j}\phi_{\lambda^{\prime },j^{\prime }}dV \end{array}$$
where $C_{ijkl}$ is the stiffness tensor, ρ the mass density, and *V* the volume of the object. The Γ matrix is block-diagonal when the shape and elastic properties are symmetric with respect to the x=0, y=0, or z=0 planes [11]. In those cases, the linear system can be turned into a few smaller linear systems requiring less computer time and resources. The xyz algorithm is well-suited for shapes for which analytic expressions for the volume integrals exist (cubes, spheres, cylinders, …). It was extended in a previous work [12] to superquadrics, which provides a large variety of shapes from octahedra to spheres and cubes, which are good approximations for the rounded shapes discussed above [6], but also to wires with cross-sections delimited by superellipses.

For the calculations, we considered materials with a strong acoustic impedance mismatch to highlight problematic cases. We chose an inorganic core (gold) and an organic shell (polystyrene). Similar nano-objects are commonly investigated experimentally and theoretically, as in ligand-covered nanoparticles [13] or nanoplatelets [14] and gold nanoplates coated with an organic layer [15]. For gold, we used isotropic parameters with mass density ρ=19.283 g/cm^3^ and sound speeds $v_{L}=3330$ m/s and $v_{T}=1250$ m/s [12]. For polystyrene, we used ρ=1.05 g/cm^3^, $v_{L}=2350$ m/s, and $v_{T}=1210$ m/s [16]. The acoustic impedance for a plane wave at normal incidence is $\rho v_{L}$; therefore, the mismatch comes mainly from the very different mass densities of both materials in that case. This large difference is expected to provide a strong impedance mismatch in non-planar geometries too.

## 3. Results

### 3.1. Superquadrics with Rigid Boundary Conditions

We first modify the RR approach presented before to calculate the vibrations of clamped superquadrics whose surface is defined by the implicit Equation (Equation 3), where $L_{i}$ is the half-length in the *i* direction and $n_{i}>0$ are the shape parameters (real numbers).(3)$${\left|\frac{x}{L_{x}}\right|}^{n_{x}}+{\left|\frac{y}{L_{y}}\right|}^{n_{y}}+{\left|\frac{z}{L_{z}}\right|}^{n_{z}}=1$$

The vibrations of such objects with the rigid boundary condition can be obtained using a basis of functions that vanish at the surface of the superquadrics. Such a basis was previously introduced for spheres in Nestoklon et al. [9] and is extended to superquadrics in Equation (Equation 4). For parallelepipeds ($n_{x,y,z}\to \infty$), Equation (Equation 5) can be used instead. The functions are of class $C^{2}$ for n≥2 and positive inside the superquadrics. They also vanish at the surface. Their integrals in Equations (Equation 1) and (Equation 2) can be computed in the first octant (x,y,z>0) [12] because they are just linear combinations of the ones in the original xyz method. Values in the other octants are obtained easily by considering the parity of *p*, *q*, and *r*. The symmetry considerations used for the free vibrations also remain valid allowing us to compute independently the frequencies for each irreducible representation.(4)$$u_{pqr}\left(x,y,z\right)=x^{p}y^{q}z^{r}\times \left(1-{\left|\frac{x}{L_{x}}\right|}^{n_{x}}-{\left|\frac{y}{L_{y}}\right|}^{n_{y}}-{\left|\frac{z}{L_{z}}\right|}^{n_{z}}\right)$$(5)$$u_{pqr}\left(x,y,z\right)=x^{p}y^{q}z^{r}\times \left(1-\frac{x^{2}}{L_{x}^{2}}\right)\left(1-\frac{y^{2}}{L_{y}^{2}}\right)\left(1-\frac{z^{2}}{L_{z}^{2}}\right)$$

As an example, Figure 1 shows the frequencies computed for superquadrics made of gold with octahedral symmetry ($L_{x}=L_{y}=L_{z}$ and $n_{x}=n_{y}=n_{z}=n$) and p+q+r≤N=20. The shape varies from a sphere when the shape parameter n=2 to a cube as n→∞. We took into account the symmetry of the problem (point group O_h_) to calculate the vibrations corresponding to each irreducible representation. Only the Raman active ones are shown: A_1g_, E_g_, and T_2g_. As with the free vibrations [12], these frequencies show only small variations when the shape varies from a sphere to a cube after normalization by multiplying with the cubic root of the volume.

The convergence of the algorithm was checked for an isotropic sphere for which analytical solutions exist (markers in Figure 1) [17]. The difference between the frequencies obtained with both methods is less than 1% up to frequency $\times \;V^{1/3}≃$ 6900 m/s. When considering all the possible irreducible representations, more than the first 750 lowest frequencies are reproduced to better than 0.1%. The frequencies for the cube using Equation (Equation 5) were also checked with FEM calculations with the FreeFem++ software (version 4.15) [18] using a mesh characteristic length 30 times smaller than the edge of the cube. A very good agreement was obtained with differences of <0.1% up to ≃4740 m/s (300 lowest frequencies considering all the irreducible representations). Overall the convergence of the RR calculations for the rigid boundary condition is as good as with the free one for spheres and cubes. We checked the convergence of the calculations using Equation (Equation 4) as $\frac{1}{n}\to 0$ toward the values calculated for the cube with Equation (Equation 5). We performed a linear extrapolation of the value obtained for $\frac{1}{n}=\frac{1}{50}$ and $\frac{1}{n}=\frac{1}{100}$ and compared it with the results discussed above. The difference was <1% below 5000 m/s demonstrating a very good convergence of this method too, even for rather large values of *n*.

The Raman active vibrations for the sphere are the spheroidal modes with ℓ=0 (A_1g_ in O_h_) and ℓ=2 (E_g_ + T_2g_ in O_h_), which are marked with disks in Figure 1. For the free vibrations, the breathing modes can also be identified by calculating the variation of the volume during oscillation. However, this method cannot be applied when the surface is clamped because the volume does not change. We also note that the method introduced here with Equation (Equation 4) is applicable when the shape parameter n>1. We only considered n≥2 in this work because the second derivative of some basis functions diverges at the origin for 1<n<2.

### 3.2. Thickness Breathing Vibration of Core–Shell Slabs

To demonstrate the issues faced with the xyz method for core–shell objects, we start with a simpler 1D system for which analytic solutions exist and displacements can be represented more easily. We consider the thickness breathing vibration of an infinite “core-shell” plate with a gold core of thickness *h* covered on both faces by layers of polystyrene with thicknesses *a* (total thickness h+2a between z=−h/2−a and z=h/2+a). We only consider vibrations at the center of the Brillouin zone $\vec{u}\left(x,y,z\right)=u\left(z\right)\vec{e_{z}}$ (vanishing wavevector) with odd longitudinal displacements u(−z)=−u(z). For the analytical solution, $u\left(z\right)=A\sin\left(q_{Au}z\right)$ inside the gold core, $u\left(z\right)=B\cos\left(q_{ps}z\right)+C\sin\left(q_{ps}z\right)$ for the upper polystyrene layer (z>h/2) and $u\left(z\right)=-B\cos\left(q_{ps}z\right)+C\sin\left(q_{ps}z\right)$ for the bottom layer (z<−h/2) with $\omega =q_{Au}v_{L,Au}=q_{ps}v_{L,ps}$. Eigenfrequencies calculated by searching the roots of the secular equation obtained from the usual continuity equations at the interfaces and the free boundary condition at the surfaces are given in Table 1 (middle). The displacements of the lowest frequency eigenmodes normalized at the surface are shown in Figure 2 (colored areas).

We implemented the xyz algorithm to solve the same problem by also considering only odd displacements with functions $u_{i}\left(z\right)=z^{2i+1}$ and replacing the volume integrals in Equations (Equation 1) and (Equation 2) by integrals from z=−h/2−a to z=h/2+a. The results are also shown in Table 1 and Figure 2 (black curves). The convergence is slow, as shown by the variations of the frequencies as the number of functions in the basis $N_{b}$ increases. The frequency of the fundamental mode deviates by more than 150 m/s (∼7%) from the analytic solution, which is large when compared with the accuracy generally for the free vibrations. The deviations for the highest overtones are very large. This slow convergence is confirmed by the corresponding displacements, which all deviate noticeably from the analytical solution. The disagreement is stronger when the change of slope at the interface is more abrupt.

To overcome this issue, we propose to supplement the original basis with $u_{i}^{\prime }$ functions, which would be suitable for the core with the rigid boundary condition and which are extended in the shell with a vanishing value as presented in Equation (Equation 6). A few $u_{i}$ and $u_{i}^{\prime }$ functions are plotted in Figure 3 after normalization at their maximum. The $u_{i}^{\prime }$ are continuous at the core–shell interface because they vanish at z=±h/2 and are differentiable inside each domain. They are linear combinations of the original $u_{i}$ functions because $u_{i}^{\prime }\left(z\right)=u_{i}\left(z\right)-u_{i+1}\left(z\right)/{\left(h/2\right)}^{2}$, but for $\left|z\right|\leq \frac{h}{2}$ only. Therefore, the integrals in Equations (Equation 1) and (Equation 2) still have analytical expressions. Finally, the parity of the $u_{i}$ and $u_{i}^{\prime }$ functions are the same, and the previous symmetry considerations remain unchanged. The key point with this choice is that the left and right derivatives at the interface are different even for small values of *i*, which is expected to help fit the analytical displacement.(6)$$u_{i}^{\prime }\left(z\right)=\left\{\begin{array}{cc} z^{2i+1}\times \left(1-{\left(\frac{z}{h/2}\right)}^{2}\right), & \;|z|\leq \frac{h}{2}\\ 0, & \;|z|\geq \frac{h}{2} \end{array}\right.$$

The frequencies and displacements obtained with the new basis (called xyz+r from now on) are also shown in Table 1 (bottom) and Figure 2 (red curves). The convergence is significantly improved when compared with the xyz basis for an identical maximum exponent value ($N_{max}=23$) or an identical number of functions in the basis ($N_{b}=12$). For the fundamental mode, the frequency deviates from the analytical one by less than $3\times 10^{-5}$ m/s compared to more than 160 m/s with xyz. In particular, we note that the first two frequencies obtained for the xyz+r method with the smallest number of functions considered in this work ($N_{b}=6$ and $N_{max}=7$) are closer to the analytical values than the frequencies calculated with xyz up to $N_{b}=12$ and $N_{max}=23$. The xyz+r displacements are also very close to the analytical ones (indistinguishable in Figure 2). The maximum difference in the normalized displacements is <$10^{-5}$ for the two lowest frequency modes, <$4\times 10^{-5}$ for the third, <$2\times 10^{-4}$ for the fourth and fifth and <$3\times 10^{-3}$ for the sixth. It is >$5\times 10^{-2}$ for all the modes calculated with xyz. Adding the rigid boundary functions to improve the convergence of the RR method is therefore very promising and will be tested in the following with 3D objects.

We note that another approach has been proposed to improve the convergence for two other 1D problems with stepped structures (longitudinal vibrations of bars and flexural vibrations of rectangular plates) by Elishakoff et al. [19]. The proposed method introduces additional rigorous contributions to the linear system involving values of the functions at the interface. Its extension to 3D objects is challenging because it would require the calculation of surface integrals. These are unlikely to have analytic expressions for superquadrics for which no such expression exists for the surface area. This is the reason why this approach has not been considered in this work.

### 3.3. 3D Core–Shell

We now apply the method presented above to improve the convergence of the xyz algorithm for 3D core–shell objects. To do so, the basis of functions is supplemented with the $u_{pqr}$ functions presented in Equation (Equation 4) inside the core, which are extended with a vanishing value inside the shell. This method was tested for a gold–polystyrene core–shell sphere using the same material parameters as before. The outer diameter is 1.2 times the inner diameter. The results are compared to analytic calculations and presented in Figure 4 for all the irreducible representations. Modes up to the second S_8_ mode at frequency×diameter≃5802.16 m/s) are shown corresponding to the 593th mode when all the degeneracies are considered.

The differences between computed and analytic values are always smaller with xyz+r, showing that the convergence is indeed improved by the additional functions. The frequency difference is less than 1% in most cases. It is noticeably larger for modes with large *ℓ* and in particular for ℓ≥9. Indeed, the number of angular oscillations increases with *ℓ*, requiring larger values of *N* to reproduce the corresponding displacement. The third S_2_ mode at 3485.73 m/s is not affected by this issue. Still, its frequency is poorly reproduced with xyz (10% error). This is problematic for a Raman-active vibration with a frequency below that of the first breathing mode (S_0_) where Raman peaks are most frequently observed. This error is reduced by more than 9 orders of magnitude with xyz+r. Similar comments apply for many other modes. Overall, xyz underperforms in most cases for this particular core–shell system. The errors are significantly larger than the ones reported above for the sphere with the rigid boundary condition. The addition of rigid functions increases significantly the convergence, resulting in errors less than $10^{-2}$ in most cases and ≤$10^{-4}$ up to the first breathing mode (S_0_).

We used the largest possible value of *N* for the xyz and xyz+r calculations in Figure 4. Increasing *N* further results in errors in the generalized eigenvalue problem solver. Overcoming this issue would require moving to a floating-point format larger than double-precision, which goes beyond the scope of this work. We used N=20 for xyz calculations and N=16 for xyz+r, resulting in a slightly larger number of functions in the xyz+r basis. To confirm that the improved convergence was not just a consequence of the larger basis, we also compared with xyz+r calculations performed with N=14. In that case, the bases were smaller for xyz+r for all the irreducible representations. The convergence was still better except for modes with large values of *ℓ* such as S_10_ and T_11_ because a lower value of *N* makes it harder to reproduce the angular oscillations in these cases.

## 4. Conclusions

To summarize, we first presented in this work an extension of the RR method to calculate the vibrations of clamped objects applicable to a large variety of shapes, from 3D superquadrics to cylinders with superellipse cross-sections. The convergence of this method is comparable to that of the original method for the free vibrations. The new functions introduced for the clamped vibrations were then shown to help improve the convergence for core–shell systems. This was first illustrated with the 1D case of the thickness breathing vibrations of “core-shell” plates. Its application to a core–shell sphere demonstrated a significantly improved convergence. The calculations presented here are applicable to a large variety of actual systems. Three-dimensional core–shell superquadrics with different parameters $\left(L_{i},n_{i}\right)$ for the core and the shell are suitable to model existing experimental inelastic light scattering results obtained for gold–silver nanorods [7]. It is also applicable to core–shell nanowires with different superellipse cross-sections for the core and the shell [12], which can model nanorods with lengths much larger than their diameters. The advantages of the RR method in terms of low computer resource requirements and symmetry are preserved, making this approach very useful to fit shape or elasticity parameters to match experimental results. 

## Figures and Tables

**Figure 1 nanomaterials-15-01865-f001:**
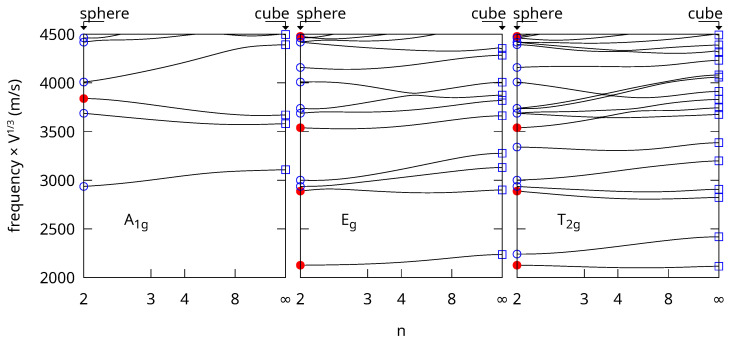
Shape variation of the A_1g_ (**left**), E_g_ (**center**) and T_2g_ (**right**) frequencies of nanoparticles made of isotropic gold with the rigid boundary condition. The frequencies are normalized after multiplication by the cubic root of the volume and are expressed in m/s. The superspheroid shape varies from a sphere (n=2) to a cube (n→∞). Normalized frequencies calculated from the analytical expressions for the sphere are plotted with round symbols. Red disks are for the spheroidal modes with ℓ=0 (A_1g_) of ℓ=2 (E_g_ + T_2g_). Other spheroidal or torsional modes are plotted with blue circles. FEM eigenfrequencies for the cube are plotted with empty blue squares.

**Figure 2 nanomaterials-15-01865-f002:**
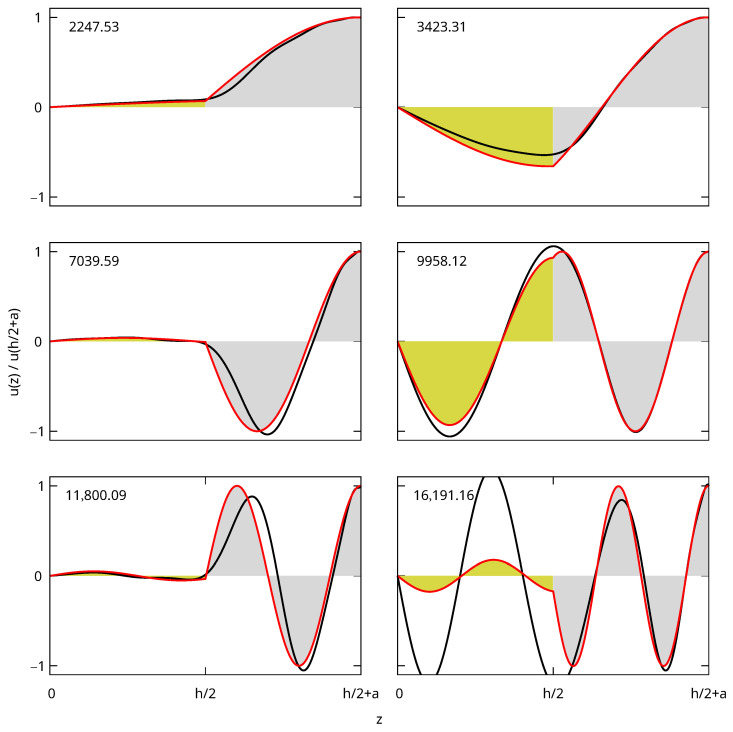
Displacements for the six eigenmodes of the core–shell slab given in Table 1. The normalized frequencies (in m/s) are indicated in the top left part of each plot. The displacements are anti-symmetrical ($u_{z}\left(-z\right)=-u_{z}\left(z\right)$) and represented for z≥0 only. The analytical solutions are shown with filled curves (yellow for gold and grey for polystyrene). The xyz displacements are plotted with a black line and the xyz+r ones in red.

**Figure 3 nanomaterials-15-01865-f003:**
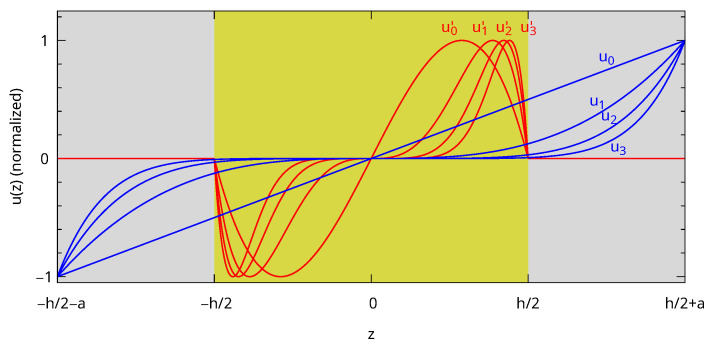
$u_{i}$ and $u_{i}^{\prime }$ functions (i=0,…,3) used in the xyz+r algorithm normalized at their maximum. The background color is yellow for gold and grey for polystyrene.

**Figure 4 nanomaterials-15-01865-f004:**
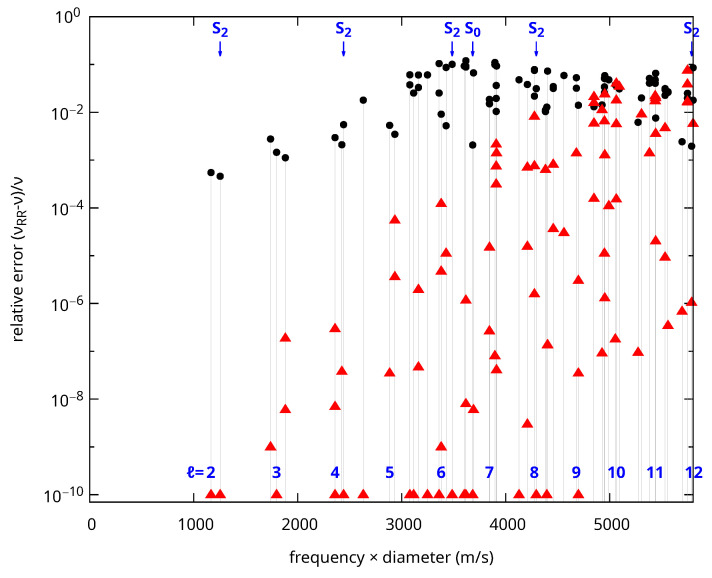
Relative error ($\left(\nu_{\mathrm{Rayleigh-Ritz}}-\nu_{\mathrm{analytic}}\right)/\nu_{\mathrm{analytic}}$) for the xyz (black disks) and xyz+r (red triangles) frequencies for a core–shell sphere with an isotropic gold core and a polystyrene shell as a function of the normalized frequency (frequency × diameter). The ratio of the shell to core diameters is 1.2. Errors below $10^{-10}$ are shown as $10^{-10}$. The positions of the Raman active spheroidal modes with ℓ=0 and 2 are indicated with arrows at the top. The position of the lowest frequency mode (torsional) for each angular momentum *ℓ* is also shown at the bottom.

**Table 1 nanomaterials-15-01865-t001:** Eigenfrequencies calculated using the analytic method and the xyz and xyz+r algorithms for the first overtones of the thickness breathing eigenmodes of an infinite core–shell plate with a gold core (thickness *h*) sandwiched between two polystyrene shell slabs (thicknesses a=h/2). The frequencies are multiplied by the total thickness h+2a and expressed in m/s. $N_{b}$: the number of functions in the basis, $N_{max}$: largest *z* exponent used in the basis. The double arrows in the first column indicate the direction for improved accuracy.

Method	$N_{b}$	$N_{\max}$	Frequency × Total Thickness (m/s)					
xyz	6	11	2608.98	3490.04	9059.21	9973.30	16,066.04	34,164.95
	7	13	2539.65	3472.34	8263.32	9969.51	15,393.52	17,773.62
	8	15	2533.52	3470.00	8259.65	9966.11	14,698.40	16,991.05
⇓	9	17	2471.79	3456.12	7927.41	9962.83	13,944.58	16,599.73
	10	19	2453.82	3453.18	7821.38	9962.68	13,356.83	16,568.04
	11	21	2445.02	3451.14	7787.45	9961.65	13,323.51	16,560.45
	12	23	2412.39	3445.44	7645.69	9961.65	12,953.16	16,556.57
analytic			2247.53	3423.31	7039.59	9958.12	11,800.09	16,191.16
xyz+r	22	23	2247.53	3423.31	7039.59	9958.12	11,800.09	16,191.26
	20	21	2247.53	3423.31	7039.59	9958.12	11,800.10	16,191.58
	18	19	2247.53	3423.31	7039.60	9958.12	11,800.10	16,196.30
	16	17	2247.53	3423.31	7039.61	9958.14	11,802.25	16,273.71
⇑	14	15	2247.54	3423.31	7039.82	9958.20	11,838.04	16,316.39
	12	13	2247.64	3423.31	7041.55	9958.96	11,890.71	16,557.73
	10	11	2248.41	3423.31	7042.05	9979.64	12,185.18	16,644.00
	8	9	2254.85	3423.44	7188.40	9991.92	16,590.86	20,576.40
	6	7	2313.28	3430.40	9743.33	11,191.98	17,406.92	43,458.30

## Data Availability

The original contributions presented in this study are included in the article. Further inquiries can be directed to the corresponding author.

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
