# Peer review of "Rayleigh–Ritz Approximation of the Acoustic Vibrations of Clamped Superquadrics—Application to Free Core–Shell Objects"

_nanomaterials, 2025, doi:10.3390/nano15241865_

Round 1

Reviewer 1 Report

Comments and Suggestions for Authors

This paper focuses on the extension and improvement of the Rayleigh-Ritz (RR) method, proposing innovative solutions to the problem of calculating acoustic vibrations of clamped superquadric objects and core-shell structures. It possesses certain academic value and application prospects. By adding new basis functions adapted to the rigid boundary conditions of the core (the xyz+r method), the paper effectively addresses the issue of poor convergence of the traditional xyz algorithm in core-shell structures. Moreover, through verification using models such as 1D core-shell slabs and 3D core-shell spheres, the effectiveness of the proposed method is confirmed, which provides reference significance for studies related to the vibrational characterization of nanomaterials.

Questions

  1. To investigate the vibration calculation of core-shell structures, the paper selects two materials with significant acoustic impedance differences. It is recommended to supplement relevant reasons and explanations, such as content related to the properties of the selected materials and the values of their impedance differences.
  2. In Section 3.2, the authors point out that Elishakoff's method is not suitable for extension to 3D cases, but insufficient analysis is provided. It is recommended to compare the two methods (the proposed xyz+r method and Elishakoff's method) in terms of computational efficiency, accuracy, and other aspects.
  3. In Section 3.3, the authors note that increasing the value of N may lead to deviations in eigenvalues. Are there any other parameters that could potentially be optimized? If not, please provide an explanation.
  4. The paper uses tables containing a large amount of data. It is recommended to reformat and organize these tables, highlight key data, and make them more accessible and easy to understand.

Author Response

Comment 1: 
This paper focuses on the extension and improvement of the Rayleigh-Ritz (RR) method, proposing innovative solutions to the problem of calculating acoustic vibrations of clamped superquadric objects and core-shell structures. It possesses certain academic value and application prospects. By adding new basis functions adapted to the rigid boundary conditions of the core (the xyz+r method), the paper effectively addresses the issue of poor convergence of the traditional xyz algorithm in core-shell structures. Moreover, through verification using models such as 1D core-shell slabs and 3D core-shell spheres, the effectiveness of the proposed method is confirmed, which provides reference significance for studies related to the vibrational characterization of nanomaterials.

Response 1:
Thank you for this positive feedback.

Comment 2:
Questions
- To investigate the vibration calculation of core-shell structures, the paper selects two materials with significant acoustic impedance differences. It is recommended to supplement relevant reasons and explanations, such as content related to the properties of the selected materials and the values of their impedance differences.

Response 2:
A short sentence was added at the end of the Methods section to highlight that both materials have very different mass densities which result in significant change of slopes at the interface. Note that the concept of acoustic impedance is not very intuitive in this case for non-planar symmetries (spherical or cubic). This was discussed in a previous work only for purely longitudinal spherical waves.
See L. Saviot and D. B. Murray, Long lived acoustic vibrational modes of an embedded nanoparticle", Phys. Rev. Lett. 93, 055506 (2004)
https://doi.org/10.1103/PhysRevLett.93.055506

Comment 3:
- In Section 3.2, the authors point out that Elishakoff's method is not suitable for extension to 3D cases, but insufficient analysis is provided. It is recommended to compare the two methods (the proposed xyz+r method and Elishakoff's method) in terms of computational efficiency, accuracy, and other aspects.

Response 3:
The Elishakoff's method has only been derived for simpler 1D cases. As written in the manuscript, implementing a similar approach for 3D objects would require the calculation of surface integrals for which no analytical expressions exist which is very problematic. For example, there is no analytical expression for the surface of a superquadrics or even the length of a superellipse.

Comment 4:
- In Section 3.3, the authors note that increasing the value of N may lead to deviations in eigenvalues. Are there any other parameters that could potentially be optimized? If not, please provide an explanation.

Response 4:
Increasing N in the Rayleigh-Ritz method is roughly equivalent to using a finer mesh in the finite-element method. It provides a better approximation of the actual displacement. This manifests as a slow decrease of the eigenfrequencies when approaching the actual solution. This is a well-known behavior. To the best of our knowledge, no other parameter can be optimized.

Comment 5:
- The paper uses tables containing a large amount of data. It is recommended to reformat and organize these tables, highlight key data, and make them more accessible and easy to understand.

Response 5:
Thank you very much for this comment which made us think of an alternative way to present these results. We replaced the tables with a single figure showing the improved convergence for all the modes. We believe that this figure is easier to understand and much less intimidating than the 3 previous tables.

Reviewer 2 Report

Comments and Suggestions for Authors

Dear authors,

the paper "Rayleigh-Ritz Approximation of the Acoustic Vibrations of Clamped Superquadrics - Application to Free Core-Shell Objects" by Sajana et al. deals with the numeric calculation of nanoparticle core-shell vibrational/acoustic eigenmode frequencies.

The use of Rayleigh-Ritz is well established in plenty numerical methods and thereby its use not new. 

The novelty might lie in the geometry, but since this is not explained well and not figures for the exact description is provided, even this chance to show novelty is not taken.

Since nano particles at first approximation are sphere the use of cartesian coordinates is surprising. Again the clarification of the exact geometry hinders to understand why this might make sense. The mentioned symetry in line 85 and 86 makes the situation even worse.

I wondered in the above mentioned lines, if the authors might have mixed symmetry with isotropy and homogenity. The reason for this is, that there are symmetric case of unisotropic materials which have off-diagonal elements in the stiffness tensor.

Concerning the material properties a reference is missing that supports the values and that they are still valid for the nano-scale, i.e. on thickness relevant for nano shell particles. I know for sure that for electric conductivies on that length scale macroscopic values are no longer ok.

If only n>2 is used, the completness of base functions that is required for Ritz theorem is no longer given. Please comment on this. (line 109)

Lines 117 ff. come fully unmotivated. It seems that some text is missing to make the link and sense to the part in before. A_1g, E_g, ... are also not explained anywhere.

Not all indices in eq 1 and 2 are explained, e.g. \lambda

n is element of the natural numbers. Hence, I have no idea why the continous lines in figure 1 make sense.

Line 126 sound to me like the FEM grid having only a resolution of 30 elements in each spacial direction. This seems  to be a low number especially  since I expect the core to be much thinner than the core radius. But this is all unclear, because the authors newer define the geometry they calculate well.

And the above reasons lead to the point that the frequencies later provided in table 1 are also useless. There is no way that any research could replicate the results of this work, due to insufficient information.

I also miss an comparison to experimental results that show that the simulation provide any results that is comparible with experimental finding. If there are no experimental results yet, at least a detailed description of possible experiments and which experimental data to compare with the simulation results here has to be provided.

Therefore I recommend rejecting this paper and to invite a resubmission of an improved version of the paper which allows replication by other researchers (i.e. sufficient detail of information) and a comparision to experimental work as well as a detailed comparison to other simulation methods.

Author Response

Comment 1:
Dear authors,

the paper "Rayleigh-Ritz Approximation of the Acoustic Vibrations of Clamped Superquadrics - Application to Free Core-Shell Objects" by Sajana et al. deals with the numeric calculation of nanoparticle core-shell vibrational/acoustic eigenmode frequencies.

The use of Rayleigh-Ritz is well established in plenty numerical methods and thereby its use not new.

Response 1:
Indeed, the Rayleigh-Ritz method is well established. However, we know of no other work dealing with the vibrations of superquadrics with the rigid boundary condition or proposing a method suitable for the free vibrations of core-shell superquadrics. These points are emphasized several times from the title of the manuscript to the conclusion and are of particular relevance today because as pointed in the manuscript experimental results concerning such systems have been obtained recently. We therefore believe that this manuscript presents new results and a timely method relevant for recently obtained experimental results.

Comment 2:
The novelty might lie in the geometry, but since this is not explained well and not figures for the exact description is provided, even this chance to show novelty is not taken.

Response 2:
The geometry of the objects considered in this work are rather simple. The first part deals with 3D shapes varying between spheres and cubes as in a previous work (superquadrics). Then we consider a 1D object consisting in a gold slab sandwiched between 2 polystyrene slabs. It is represented in Figure 3. The last system is just a 3D spherical core-shell. These points are clearly stated in the text.

Comment 3:
Since nano particles at first approximation are sphere the use of cartesian coordinates is surprising. Again the clarification of the exact geometry hinders to understand why this might make sense. The mentioned symmetry in line 85 and 86 makes the situation even worse.

Response 3:
We agree that the use of Cartesian coordinates is at first surprising for systems with the spherical symmetry. However, it has been shown multiple times to be very efficient nevertheless, including in the original work by Visscher et al. In addition, spherical systems are considered in this work only because analytical solutions exists which is of great help to check the convergence of the method. One goal of this work is to provide a method suitable for nanocubes made of cubic materials (gold, silver). In that case the symmetry is cubic (Oh) and the use of Cartesian coordinates is appropriate.

Comment 4:
I wondered in the above mentioned lines, if the authors might have mixed symmetry with isotropy and homogenity. The reason for this is, that there are symmetric case of unisotropic materials which have off-diagonal elements in the stiffness tensor.

Response 4:
The reviewer is right that anisotropy may come from the stiffness tensor alone even for an object with a spherical geometry. The physical system has the spherical symmetry only for spherical shapes and isotropic elasticity. This is the case for the calculations with the sphere with the rigid boundary condition and the core-shell because isotropic elasticity is considered in both cases in this work. A sphere made of a monodomain cubic material or a superquadrics or a cube have cubic symmetry (Oh). These are the kind of objects which are ultimately interesting (gold silver coreshell superquadrics for which recent experimental results have been obtained).

Comment 5:
Concerning the material properties a reference is missing that supports the values and that they are still valid for the nano-scale, i.e., on thickness relevant for nano shell particles. I know for sure that for electric conductivies on that length scale macroscopic values are no longer ok.

Response 5:
We added references for the values of the mass densities and sound speeds. We agree with the reviewer that the properties of a nanoscale object may be different from those of the bulk material. However, we do not try to interpret actual experimental values in this work. In addition, the systems of interest have dimensions of a few tens of nanometers whose elastic properties do not deviate significantly from those of the bulk. Finally, no actual size is used in the manuscript as we present results with normalized frequencies (frequency multiplied by diameter).

Comment 6:
If only n>2 is used, the completness of base functions that is required for Ritz theorem is no longer given. Please comment on this. (line 109)

Response 6:
Lowercase "n" is a shape parameter. It is a real number. n=2 for spheres and the shape transforms into a cube as n tends to infinity.
Uppercase "N" is indeed an integer and represent the largest exponent sum (p+q+r<=N) in the basis functions. We modified the text to emphasize that "n" is a shape parameter in a few places to help the reader distinguish between n and N.

Comment 7:
Lines 117 ff. come fully unmotivated. It seems that some text is missing to make the link and sense to the part in before. A_1g, E_g,
... are also not explained anywhere.

Response 7:
Thank you for this comment. We have modified this part to introduce the point group and irreducible representations more clearly.

Comment 8:
Not all indices in eq 1 and 2 are explained, e.g. \lambda

Response 8:
Thank you for pointing out this deficiency. We modified the text to explain that lambda represents (p,q,r) and also added a sentence to redirect the reader to the original work for a comprehensive presentation of the xyz method.

Comment 9:
n is element of the natural numbers. Hence, I have no idea why the continous lines in figure 1 make sense.

Response 9:
See previous reply about the difference between lowercase "n" and uppercase "N".

Comment 10:
Line 126 sound to me like the FEM grid having only a resolution of 30 elements in each spacial direction. This seems to be a low number especially since I expect the core to be much thinner than the core radius. But this is all unclear, because the authors newer define the geometry they calculate well.

Response 10:
Line 126 is about a cube made of gold, i.e., without a shell. The number of nodes in about 30^3 and the number of tetrahedra is larger (about 120000). The computer resources required to perform the calculations are much larger than the ones required for the Rayleigh-Ritz calculation both in terms of CPU time and RAM without providing any direct quantitative information about the symmetry of the vibrations. The FEM calculation were designed to confirm the validity of the Rayleigh-Ritz calculations in the absence of similar calculations in the literature. The very small difference obtained between the FEM and RR frequencies confirms this validity.

Comment 11:
And the above reasons lead to the point that the frequencies later
provided in table 1 are also useless. There is no way that any
research could replicate the results of this work, due to insufficient
information.

Response 11:
We fail to see which information is missing. The geometry is shown in Figure 3 (and 2). The analytic expressions are given in the text (line 146-149). The numerical approach is presented in the text and in particular through Equations 1-6. The mass densities and sound speeds of the materials are given in the Methods section. The dimensions are also in the caption of Table 1 (a=h/2).

Comment 12:
I also miss an comparison to experimental results that show that the simulation provide any results that is comparible with experimental finding. If there are no experimental results yet, at least a detailed description of possible experiments and which experimental data to compare with the simulation results here has to be provided.

Response 12:
We agree with the reviewer that this work is of interest only because it helps to understand experimental results. As stated above, this is clearly stated throughout the text and in particular in the introduction and conclusion. However, we believe that the methods for the calculation of the vibrations with the rigid boundary condition and its application to improve the convergence for core-shell objects are new and general enough to warrant a dedicated presentation. In addition, the presentation and discussion of the experimental results is not straightforward. A manuscript mixing both aspects would be too long and hard to follow.

Comment 13:
Therefore I recommend rejecting this paper and to invite a resubmission of an improved version of the paper which allows replication by other researchers (i.e. sufficient detail of information) and a comparision to experimental work as well as a detailed comparison to other simulation methods.

Response 13:
We have replied to and taken into account all the comments of the two reviewers. The modifications introduced in the new version of the manuscript are significant (Tables replaced by a Figure) and should make it much easier to follow. We have also addressed all the criticisms. We believe that the manuscript now meets all the requirements to be accepted for publication in Nanomaterials.

Round 2

Reviewer 1 Report

Comments and Suggestions for Authors

The authors have answered all of my questions. It can be accepted for publication.